# Dach1 is essential for maintaining normal mature podocytes

Keiko Tanaka[1,2], Haruko Hayasaka[3], Taiji Matsusaka[1]*

1 Departments of Physiology, Tokai University School of Medicine, Kanagawa, Japan, 2 Department of Nephrology, Rheumatology, Endocrinology and Metabolism, Okayama University Faculty of Medicine, Dentistry and Pharmaceutical Sciences, Okayama, Japan, 3 Department of Science, Faculty of Science & Engineering, Graduate School of Science and Engineering, Kindai University, Osaka, Japan

* taijim@is.icc.u-tokai.ac.jp

**Data Availability Statement:** All relevant data are within the paper and its Supporting information files.

**Funding:** This study was supported by the following means: Japan Society for the Promotion of Science, 21H02936, awarded to TM.

## Abstract

Dach1 is highly expressed in normal podocytes, but this expression rapidly disappears after podocyte injury. To investigate the role of Dach1 in podocytes *in vivo*, we analyzed global, podocyte-specific, and inducible *Dach1* knockout mice. Global Dach1 knockout (*Dach1⁻/⁻*) mice were assessed immediately after birth because they die within a day. The kidneys of *Dach1⁻/⁻* mice were slightly smaller than those of control mice but maintained a normal structure and normal podocyte phenotypes, including ultrastructure. To study the role of Dach1 in mature podocytes, we generated *Dach1* knockout mice by mating *Dach1^fl/fl^* mice with *Nphs1*-Cre or *ROSA*-CreERT2 mice. Due to inefficient Cre recombination, only a small number of podocytes lacked Dach1 staining in these mice. However, all eleven *Nphs1*-Cre/*Dach1^fl/fl^* mice displayed abnormal albuminuria, and seven (63%) of them developed focal segmental glomerulosclerosis. Among 13 *ROSA*-CreERT2/*Dach1^fl/fl^* mice, eight (61%) exhibited abnormal albuminuria after treatment with tamoxifen, and five (38%) developed early sclerotic lesions. These results indicate that while Dach1 does not determine the fate of differentiation into podocytes, it is indispensable for maintaining the normal integrity of mature podocytes.

## Introduction

Podocytes are terminally differentiated glomerular cells that play a vital role in the filtration barrier. Podocyte injury leads to albuminuria, loss of podocyte-specific markers, and the development of segmental glomerulosclerosis. The development and maintenance of podocytes are dependent on a complicated network of transcription factors, including WT1, MAFB, TCF21, LMX1B, and FOXC2 [1, 2].

Dachshund homolog 1 (Dach1) is a transcription factor that determines cell fate in various organs, such as the eye, leg, and nervous system [3, 4]. Null mutations of the dachshund gene in flies cause anophthalmia and shortened legs. Dach1 also functions as a tumor suppressor, inhibiting cell growth and migration by directly repressing estrogen-α receptor and androgen receptor signaling [5, 6]. It regulates the cell cycle and inhibits cell proliferation by interacting with cyclin D1, p53, c-Jun, YB-1, and SMAD4 in several types of cancer cells [7–12].

**Competing interests:** The authors have declared that no competing interests exist.

Dach1 participates in kidney development as a member of the *Eya-Six-Dach* network [13, 14]. Six1 and Eya1 initiate ureteric budding by transcriptional activation of glial cell line-derived neurotrophic factor (Gdnf), resulting in branching of the ureter via Ret and Gfra1. Dach1 functions as a corepressor or coactivator of Six1, with Eya1 in the network [15]. *DACH1* has been proposed as a candidate gene for renal hypoplasia, characterized by small or disorganized kidneys following abnormal organogenesis [16]. Double-homozygous missense mutations of *DACH1* and *BMP4* were found in a patient with bilateral renal cystic dysplasia [15]. *DACH1* may be involved in the pathogenesis of branchio-oto-renal (BOR) syndrome [17], which comprises numerous congenital anomalies, including branchial arch deformation, hearing loss, and variable renal anomaly.

Dach1 is expressed in podocytes and tubular epithelial cells of adult and developing kidneys in both humans and mice [18]. *DACH1* polymorphisms have been reported to be associated with nephrotic syndrome and chronic kidney diseases [19–21]. It has been reported that DACH1 is downregulated in the renal tissue of patients with glomerulopathies, such as diabetic nephropathy, IgA nephropathy, idiopathic membranous nephropathy, and minimal change disease, compared to healthy control individuals [2, 22]. We and others previously reported that Dach1 is highly expressed in normal podocytes, but this expression rapidly disappears after the induction of podocyte injury [2, 22, 23]. This pattern is similar to that of other proteins essential for podocytes, such as WT1. These indicate that Dach1 is crucial for normal podocytes similarly to WT1, although a direct association between the two factors was not reported.

RJ Davis et al. reported the first study of *Dach1* knockout (KO) mice. They found that homozygous mutant mice postnatally died within two days; these mice exhibited failure to suckle, cyanosis, and respiratory distress. No histological abnormality was found in the kidney, and serum blood urea nitrogen, calcium, creatinine, potassium, and phosphate levels were normal [24, 25].

However, Cao et al. reported that global *Dach1* null-mutant mice showed severe renal hypoplasia, immature podocytes, and absence of foot-process formation. In addition, they generated podocyte-specific *Dach1* KO and overexpressing mice. Podocyte-specific *Dach1* KO mice showed a normal renal phenotype at baseline but developed severe proteinuria and podocyte injury after treatment with low-dose streptozotocin or adriamycin. Overexpression of Dach1 in podocytes of diabetic OVE26 mice attenuated podocyte injury [26].

Doke et al. generated distal tubule-specific *Dach1* KO mice. These mice were healthy and showed normal kidney phenotypes at baseline. After injection with folic acid or injection with streptozotocin and heminephrectomy, *Dach1* KO mice showed severe renal fibrosis. In contrast, distal tubule-specific Dach1 overexpression attenuated renal fibrosis induced by folic acid injection [27].

In our study, we independently generated podocyte-specific and inducible *Dach1* KO mice. Unlike the phenotypes reported by Cao, our podocyte-specific *Dach1* KO mice showed proteinuria at baseline. Here, we report detailed analyses of the kidneys of global *Dach1* KO mice, podocyte-specific *Dach1* KO mice, and inducible *Dach1* KO mice.

## Materials and methods

### Animal ethics

All animal experiments were approved by the Animal Experimentation Committee of Tokai University School of Medicine. All animal experiments were performed in accordance with relevant guidelines and regulations, and the study is reported in accordance with ARRIVE guidelines (https://arriveguidelines.org).

## Mice

The *Dach1* null mouse (*Dach1⁻*) line (Dach1<tm1.1Hhsk>, deposited in Riken BRC as RBRC06804) and *Dach1ᶠˡ* mouse line (Dach1<tm1.2Hhsk>, RBRC10097) were generated by a targeted mutation in KY1.1 ES cells, which were derived from (C57BL/6J x 129S6/SvEvTac) F1 mice. *Dach1⁻/⁻* mice with the exon 1 deletion were generated by mating heterozygous *Dach1⁻/⁺* mice. Eight *Dach1⁻/⁻* mice, 9 *Dach1⁻/⁺* and 4 *Dach1⁺/⁺* mice were histologically analyzed at P0. Longitudinal size was measured in coronal sections (n = 3, *Dach1⁻/⁻*; n = 5, control).

The *Dach1ᶠˡ* line has two loxP sites flanking exon 1 of the *Dach1* gene [28]. To disrupt *Dach1* in podocytes, the *Dach1ᶠˡ* line was crossed with the *Nphs1*-Cre line [29], generating *Nphs1*-Cre/*Dach1ᶠˡ/ᶠˡ* mice. Both lines are on the C57BL/6 genetic background. Eleven *Nphs1*-Cre/*Dach1ᶠˡ/ᶠˡ* mice (8 males, 3 females) were analyzed for histology at 8 weeks (n = 3), 5 months (n = 1), or 7–8 months (n = 7) of age. *Nphs1*-Cre/*Dach1ᶠˡ/ᵂᵀ* mice (n = 12, 8 males, 4 females) were similarly analyzed as controls at 8–9 weeks (n = 4) or 7–8 months (n = 8) of age. Cre-mediated deletion of the *Dach1* gene was detected by PCR using primers, 5'-TGC GCT CGC TCT TTC TTA ACC TC-3' and 5'-TGT CCG GAA CGG GTG CAG GTC ACC-3'.

To examine the effect of *Dach1* deletion in adult mice, the *Dach1ᶠˡ* line was crossed with the *ROSA26*-CreERT2 (*ROSA*-CreERT2) line [30], which was on a mixed genetic background including C57BL/6 129/SvJ, and Black Swiss. Thirteen *ROSA*-CreERT2/*Dach1ᶠˡ/ᶠˡ* mice (6 males, 7 females) were treated with tamoxifen (0.1 mg/g body weight, 5 consecutive days, 3 courses), intraperitoneally (n = 1) or orally (n = 12), beginning at ages ranging from 4 to 11 weeks. Similarly treated *Dach1ᶠˡ/ᶠˡ* mice without *ROSA*-CreERT2 (3 males, 3 females) were used as controls. The urine and histology of *ROSA*-CreERT2/*Dach1ᶠˡ/ᶠˡ* mice were analyzed 3–5 days after the completion of three courses of tamoxifen treatment, at the ages of 15 weeks (n = 1), 18 weeks (n = 2) or 20–22 weeks (n = 10). Control mice were similarly analyzed at the ages of 18 weeks (n = 1) or 20–22 weeks (n = 5).

## Urinalysis

Twenty-four-hour urine was collected from 14 *Nphs1*-Cre/*Dach1ᶠˡ/ᶠˡ* and 7 control mice at 8–35 weeks of age and from 13 *ROSA*-CreERT2/*Dach1ᶠˡ/ᶠˡ* and 6 control mice at 16–27 weeks of age. Concentrations of albumin and creatinine were determined by turbidimetric immuno-assay and enzymatic methods, respectively, in an outside laboratory (SRL). In addition, some urine samples were analyzed by polyacrylamide gel electrophoresis (SDS–PAGE) to detect abnormal albuminuria.

## Histology and immunohistochemistry

Kidneys were fixed in 4% buffered paraformaldehyde and embedded in paraffin. Paraffin sections (2 μm) were stained with periodic acid-Schiff (PAS) or used for immunostaining. Information about the primary antibodies used in the study, the antigen retrieval method, and dilution ratios are listed in S1 Table. Antigen retrieval was performed by heating in citrate buffer (0.1 mol/L, pH 6.0) with a microwave or autoclave. CanGet signal solutions (Toyobo, Osaka, Japan) were used for Dach1 and WT1. For double immunostaining, the signal of the first antibody (anti-synaptopodin) was visualized by diaminobenzidine (DAB), and the slides were washed in 0.1 M glycine buffer (pH 2.2) for 30 minutes, 3 times. Then, the slides were stained with the second antibody (Dach1 or WT1) and visualized by Ni, Co-DAB. Information regarding antibodies is shown in S1 Table.

### Evaluation of glomerular size in neonatal mice

To estimate the maturity of glomeruli in neonatal mice, the diameters of glomeruli were measured in sections stained with PAS. Approximately 20–30 glomeruli in the juxtamedullary cortex were evaluated.

### Electron microscopic analyses

Kidney samples of newborn $Dach1^{-/-}$ and control mice were immersion-fixed in 4% paraformaldehyde overnight and then transferred into 2.5% glutaraldehyde for 30 min. Subsequent preparation for scanning electron microscopy (SEM) and transmission electron microscopy (TEM) was performed by standard methods. Some paraffin blocks were cut into 10 μm-sections, deparaffinized, and subjected to SEM analysis.

### Kidney biopsy

Under anesthesia with pentobarbital (50 mg/kg, i.p.) and buprenorphine (0.05 mg/kg, s.c.), the right kidney was exposed through a dorsal incision. Approximately 1 mm of the upper pole of the kidney was excised with scissors and immediately fixed in 4% buffered paraformaldehyde. The resected surface of the kidney was promptly compressed and cauterized to halt bleeding, followed by the application of a hemostatic agent. Upon confirming hemostasis, the kidney was returned to its original position, and the fascia and skin were sutured.

### Statistical analysis

The statistical analyses were performed using the JMP software program (version 11, SAS Institute Inc.; Cary, NC, USA). Data on the urinary albumin/creatinine ratio were logarithmically transformed and analyzed by unpaired t-tests. The data are represented by the geometric mean and 95% confidence interval. For other data, the results are expressed as the median and interquartile range (IQR). Differences between groups were analyzed using the Mann–Whitney U-test for continuous data. $P$ values of $<0.05$ were considered to indicate statistical significance.

## Results

### Normal kidney development in global *Dach1* KO (*Dach1$^{-/-}$*) mice

Immunostaining of wild-type newborn mouse kidneys revealed intense staining of Dach1 in the ureteric bud and faint staining in the cap mesenchyme. As the cap mesenchyme differentiated into the renal vesicle, Dach1 staining became more pronounced. Within the S-shaped bodies, both the inner thick and outer thin layers exhibited intense staining. In glomeruli at the capillary stage, Dach1 staining was most prominent in podocytes and less intense in Bowman's capsule (Fig 1A). Dach1 staining was also observed in the thick ascending limb (TAL), distal convoluted tubule (DCT), connecting tubule (CNT), and collecting duct (CD).

  $Dach1^{-/-}$ mice exhibited grossly normal kidney structure, although the size of their kidneys was smaller than that of controls (Fig 1A–1C). The glomerular size and density in $Dach1^{-/-}$ mice were similar to those in controls (Fig 1D and 1E). These results suggest that loss of Dach1 may attenuate the branching of the ureteric bud. In $Dach1^{-/-}$ mice, nuclear Dach1 staining was mostly absent, although faint staining in the ureteric bud and rare podocytes was still detectable. No differences were observed in the early nephron structure, as depicted by Six2 and WT1 staining (Fig 1A). Furthermore, no differences were observed in the immunostaining of other podocyte proteins, including nephrin, podocin, podocalyxin, synaptopodin, and nestin, between $Dach1^{-/-}$ and control mice (S1 Fig). Immunostaining for albumin indicated a similar

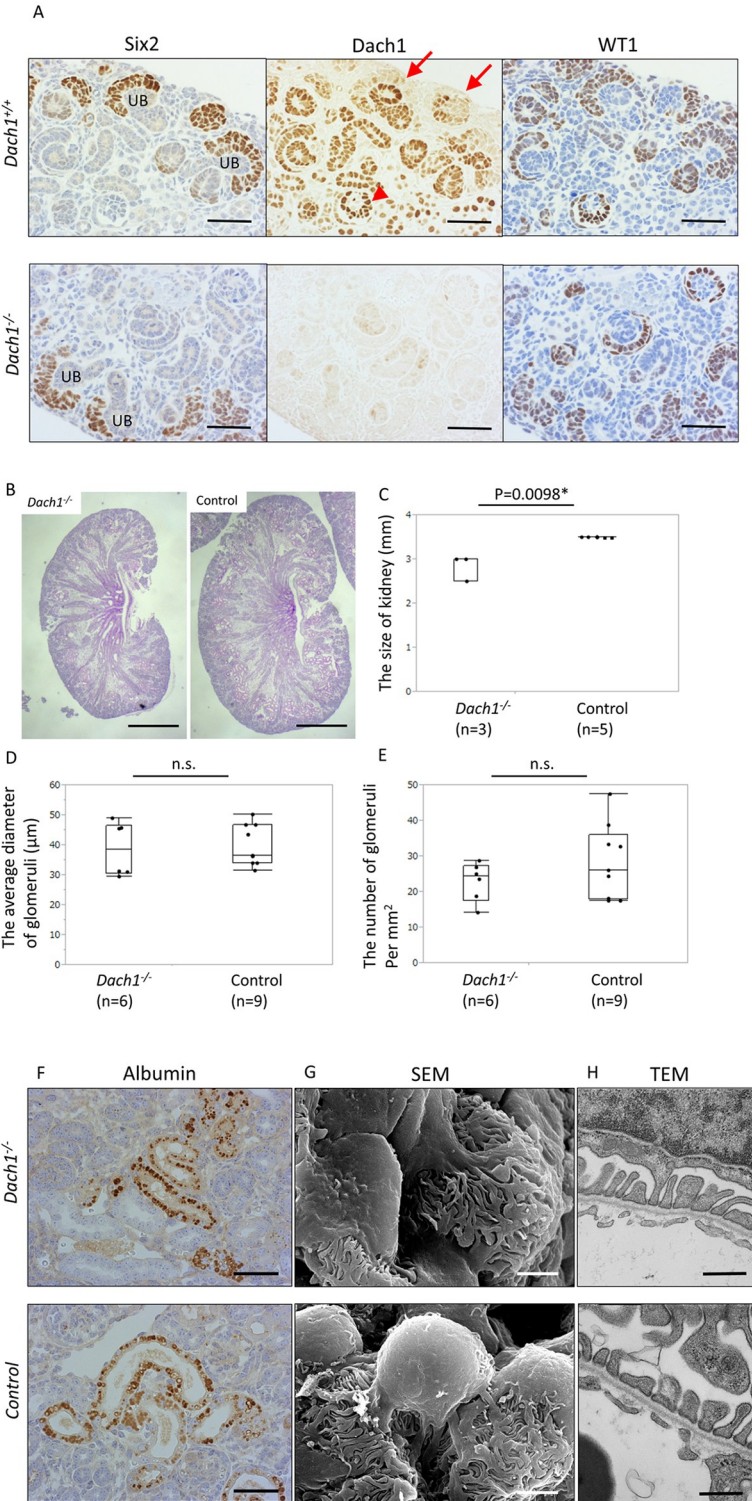

**Fig 1. Phenotypes of neonatal global *Dach1* KO (*Dach1*$^{-/-}$) mice.** (A) Immunostaining for Six2, Dach1, and WT1 in serial sections of *Dach1*$^{-/-}$ and littermate wild-type (*Dach1*$^{+/+}$) mice. In *Dach1*$^{+/+}$ mice, Dach1 is intensely expressed in the ureteric bud (UB) and podocytes (arrowheads) and faintly expressed in the cap mesenchyme (arrows). Dach1 is also expressed in developing distal tubules. *Dach1*$^{-/-}$ mice exhibit almost no Dach1 staining but maintain normal architecture. Scale bar: 50 μm. (B) Representative images of the whole kidneys. Scale bar: 500 μm. (C) Kidney longitudinal size. (D) Glomerular size. Not significant (n.s.) (E) Glomerular density. Not significant (n.s.). *Dach1*$^{-/-}$

mice had slightly smaller kidneys than the control mice ($Dach1^{-/+}$ and $Dach1^{+/+}$ mice) but similar size and density of glomeruli. (F) Immunostaining for albumin showed that $Dach1^{-/-}$ mice had a similar amount of albumin reabsorption droplets in the proximal tubules to control mice. Scale bar: 50 μm. (G) Scanning electron microscopy (SEM) (Scale bar: 2 μm) and (H) transmission electron microscopy (TEM) (Scale bar: 500 nm) showed that $Dach1^{-/-}$ mice had normal foot processes and normal slit diaphragms.

amount of albumin reabsorption in the proximal tubules of both $Dach1^{-/-}$ and control mice (Fig 1F).

SEM and TEM images revealed that the foot processes in $Dach1^{-/-}$ mice displayed a normal structure, similar to that of control mice (Fig 1G and 1H). TEM images also showed intact slit diaphragms in $Dach1^{-/-}$ mice.

These findings collectively indicate that the absence of Dach1 leads to slight hypoplasia of the kidney but does not affect the differentiation of renal cells, including podocytes, during development.

## Podocyte injury in $Nphs1$-Cre/$Dach1^{fl/fl}$ mice

Because $Dach1^{-/-}$ mice do not survive more than one day, we generated podocyte-specific $Dach1$ KO mice by mating $Nphs1$-Cre and $Dach1^{fl/fl}$ lines to assess the impact of $Dach1$ deletion on mature podocytes. To confirm Cre-mediated recombination in the glomerulus, DNA was extracted from the glomeruli and tail and subjected to PCR analysis using primers specific to the deleted $Dach1$ alleles. This analysis confirmed the deletion of the $Dach1$ allele (350 bp) in the glomeruli, while the tail did not exhibit the deletion (Fig 2A).

$Nphs1$-Cre/$Dach1^{fl/fl}$ mice were born in accordance with Mendelian inheritance ratios and exhibited grossly normal characteristics. However, all $Nphs1$-Cre/$Dach1^{fl/fl}$ mice displayed abnormal albuminuria to varying degrees. The average urinary albumin/creatinine ratio in $Nphs1$-Cre/$Dach1^{fl/fl}$ mice was 4.75 (95% CI: 1.19–8.32) mg/mgCr, whereas that of control mice was 0.071 (0.054–0.089) mg/mgCr ($P<0.0001$) (Fig 2B). At 7–8 months of age, these values were 5.82 (95% CI: 0.39–11.2) and 0.063 (0.047–0.080), respectively (S2 Fig). The proteinuria worsened as they aged (Fig 2C and S3 Fig).

In total, seven out of 11 (63%) $Nphs1$-Cre/$Dach1^{fl/fl}$ mice showed focal segmental glomerulosclerosis (FSGS) with diminished nephrin staining (Fig 2D). At 7–8 months of age, five out of 7 (71%) $Nphs1$-Cre/$Dach1^{fl/fl}$ mice showed FSGS lesions. In these five mice, sclerotic lesions were observed in 4.2% (1.0–5.8%) of the total glomeruli. The percentage of glomerulosclerosis was positively correlated with the level of albuminuria (Fig 2E). FSGS lesions were accompanied by tubulointerstitial injuries, including tubular dilation, epithelial cell damage, and the formation of albuminous casts. Thirty-six percent (4 out of 11) of $Nphs1$-Cre/$Dach1^{fl/fl}$ mice did not show FSGS or tubulointerstitial injury but exhibited increased albumin reabsorption droplets in the proximal tubules.

Double immunostaining for Dach1 and synaptopodin revealed the absence of Dach1 staining in some podocytes of nonsclerotic glomeruli in $Nphs1$-Cre/$Dach1^{fl/fl}$ mice (Fig 2F). However, the rate of $Dach1$ deletion was relatively low, with fewer than 10% of podocytes in each glomerular section exhibiting $Dach1$ deletion, indicating inefficient Cre-mediated recombination. In nonsclerotic glomeruli, $Dach1$-deleted podocytes showed normal staining for synaptopodin and WT1 in the adjacent sections (Fig 2F). The low percentage of $Dach1$ KO podocytes raised the possibility that $Dach1$ KO podocytes were lost without causing injury. To test this possibility, we counted the number of podocytes in the nonsclerotic glomerulus at 7–8 months of age. There was no difference in the number of podocytes per glomerulus between $Nphs1$-

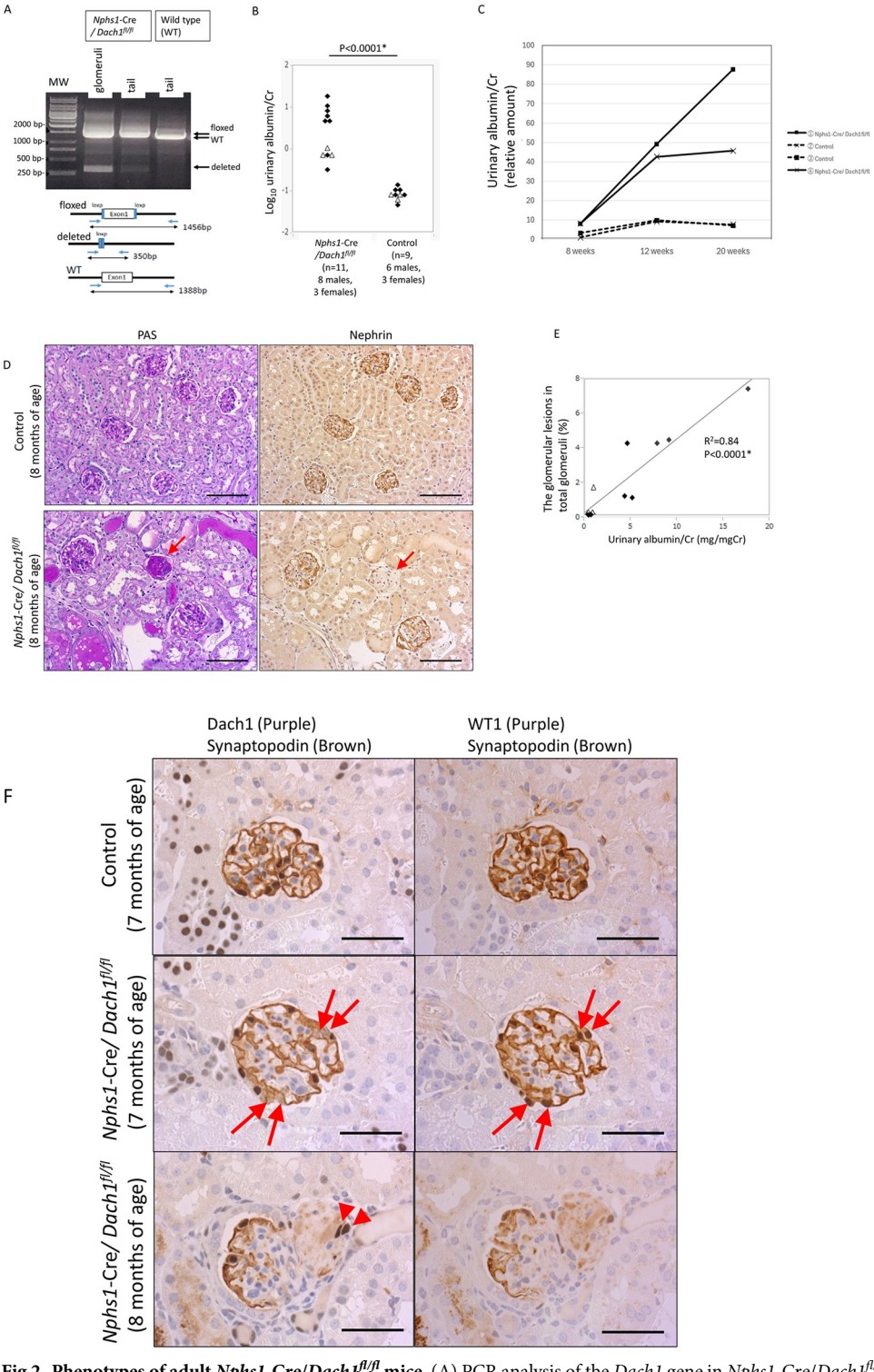

**Fig 2. Phenotypes of adult *Nphs1*-Cre/*Dach1*^fl/fl^ mice.** (A) PCR analysis of the *Dach1* gene in *Nphs1*-Cre/*Dach1*^fl/fl^ mice. Glomeruli, but not tails, contained the *Dach1* deleted allele (350 bp) in addition to the floxed allele (1456 bp). (B) Urinary albumin/creatinine (mg/mg) ratio in mice of all ages. *Nphs1*-Cre/*Dach1*^fl/fl^ mice were analyzed at 8 weeks (n = 3), 5 months (n = 1), or 7–8 months (n = 7) of age. The control mice were analyzed at 8–9 weeks (n = 3) or 7–8 months (n = 6) of age. The urinary albumin/creatinine ratio in 7- to 8-month-old mice is shown in S2 Fig. Males are represented by black diamonds, while females are represented by white triangles. (C) Urinary albumin/creatinine ratio in *Nphs1*-Cre/*Dach1*^fl/fl^ and control mice. All mice were male. The relative albumin concentration was measured by

densitometry in Coomassie blue-stained SDS gels. The number of *Nphs1*-Cre/*Dach1*$^{fl/fl}$ mice increased with age. The SDS–PAGE images are shown in S3 Fig. (D) PAS and nephrin staining in serial sections. *Nphs1*-Cre/*Dach1*$^{fl/fl}$ mice exhibit glomerulosclerosis with diminished nephrin staining (arrows). Scale bar: 100 μm. (E) Correlation of the urinary albumin/creatinine ratio and percentage of glomerular lesions in *Nphs1*-Cre/*Dach1*$^{fl/fl}$ mice of all ages (n = 11, 8 males, 3 females). 8 weeks (n = 3), 5 months (n = 1), or 7–8 months (n = 7) of age. Males: black diamonds. Females: white triangles. (F) Partial deletion of *Dach1* in the podocytes of *Nphs1*-Cre/*Dach1*$^{fl/fl}$ mice. Serial sections were doubly stained for Dach1 + synaptopodin or WT1 + synaptopodin. Some hematoxylin+ nuclei of podocytes in *Nphs1*-Cre/*Dach1*$^{fl/fl}$ mice lacked Dach1 staining (arrows) but retained synaptopodin and WT1 staining (arrows) in adjacent sections. In the sclerotic lesions of *Nphs1*-Cre/*Dach1*$^{fl/fl}$ mice (bottom), staining of Dach1, WT1, and synaptopodin were all diminished. Occasionally, a few Dach1-expressing podocytes were involved in sclerotic lesions (arrowheads). Scale bar: 50 μm.

Cre/*Dach1*$^{fl/fl}$ and control mice (S4 Fig), ruling out the possibility of a loss of *Dach1*-deleted podocytes without injury. In sclerotic glomeruli, Dach1 staining was diminished, similar to other podocyte marker proteins, such as synaptopodin, WT1 (Fig 2F) and nephrin (Fig 2D). Some injured podocytes with diminished WT1 and synaptopodin staining were found to lack Dach1 staining. Dach1-positive podocytes were also secondarily involved in sclerotic lesions (Fig 2F). SEM images showed that the foot processes of *Nphs1*-Cre/*Dach1*$^{fl/fl}$ mice were thicker than those of control mice (S5 Fig).

These findings indicated that *Dach1* deletion alone leads to injury or loss of podocytes, ultimately resulting in the development of FSGS.

## Podocyte injury in *ROSA*-CreERT2/*Dach1*$^{fl/fl}$ mice

We examined the impact of *Dach1* deletion in adult mice. 3–5 days after the completion of the 15-day tamoxifen treatment, abnormal albuminuria was observed in 8 out of 13 (61%) *ROSA*-CreERT2/*Dach1*$^{fl/fl}$ mice. The average urinary albumin/creatinine ratio in *ROSA*-CreERT2/*Dach1*$^{fl/fl}$ mice was 1.03 (95% CI: 0.23–4.57) mg/mg Cr, whereas that of control mice was 0.10 (0.039–0.18) mg/mg Cr (P<0.05) (Fig 3A).

Podocyte injury was observed in five out of 13 *ROSA*-CreERT2/*Dach1*$^{fl/fl}$ mice. One mouse exhibited moderate FSGS with focal tubular dilatation (Fig 3B), while four mice showed mild segmental sclerosis in a few glomeruli. Nephrin staining was diminished in the sclerotic lesions. Renal biopsy was performed on an additional *ROSA*-CreERT2/*Dach1*$^{fl/fl}$ mouse and a control mouse one week after the end of tamoxifen treatment. The biopsy sample showed normal renal histology with normal nephrin staining. However, 8 weeks later, the autopsy of the same mouse displayed FSGS with diminished nephrin staining (S6 Fig).

In adult control mice, Dach1 staining was observed in the podocytes, TAL, DCT, CNT, and cortical and medullary CD (CCD, MCD). In tamoxifen-treated *ROSA*-CreERT2/*Dach1*$^{fl/fl}$ mice, tubular Dach1 staining was mostly absent (Fig 3C). However, a small number of podocytes were found to lack Dach1 staining, again indicating inefficient Cre-mediated recombination in podocytes. As observed in *Nphs1*-Cre/*Dach1*$^{fl/fl}$ mice, *Dach1*-deleted podocytes showed normal staining for synaptopodin and WT1 in nonsclerotic glomeruli. Dach1-negative podocytes were found in injured glomeruli, and Dach1-positive podocytes were also involved in sclerotic lesions. SEM images again showed that foot processes were thicker in *ROSA*-CreERT2/*Dach1*$^{fl/fl}$ mice than those in control mice (S7 Fig).

Although tubular Dach1 was almost completely lost in *ROSA*-CreERT2/*Dach1*$^{fl/fl}$ mice, no abnormalities were observed in tubular structures or in the intensity and pattern of various tubular marker proteins, including uromodulin (expressed in TAL), NCC (DCT), Calbindin-D-28K (DCT and CNT), ENaCs (CNT, CCD), AQP2 (CCD, MCD) and E-Cadherin (TAL-MCD) (S8 Fig).

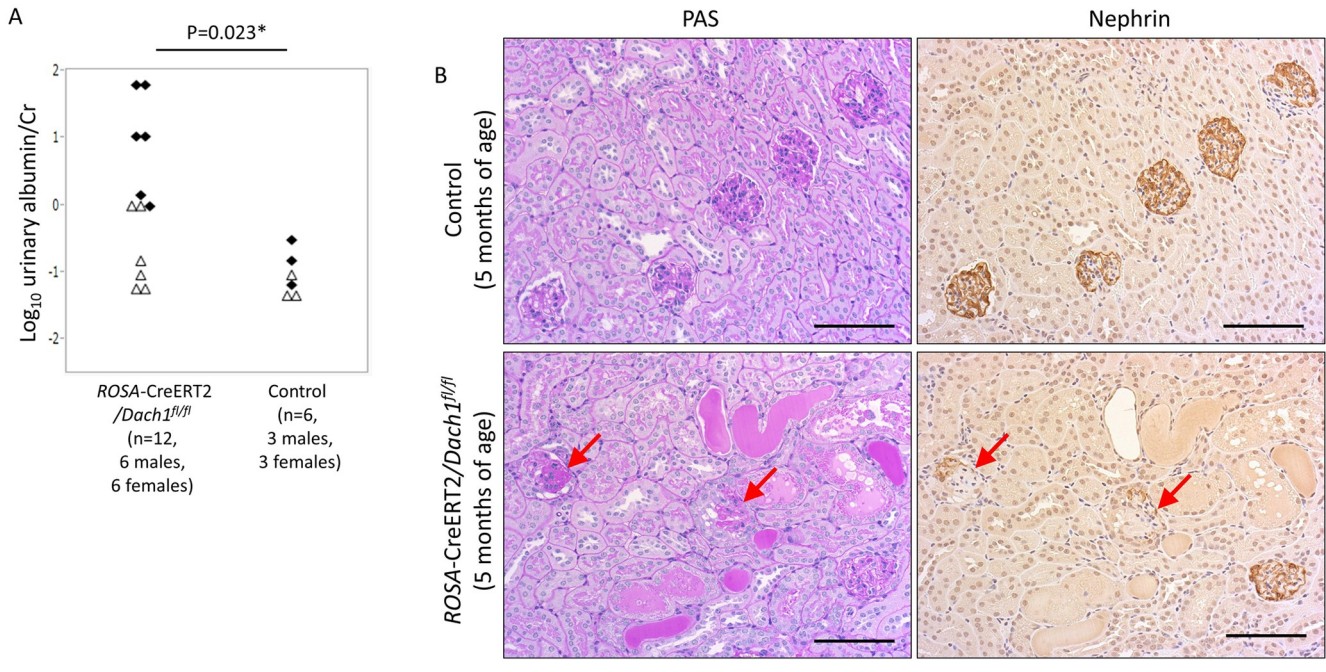

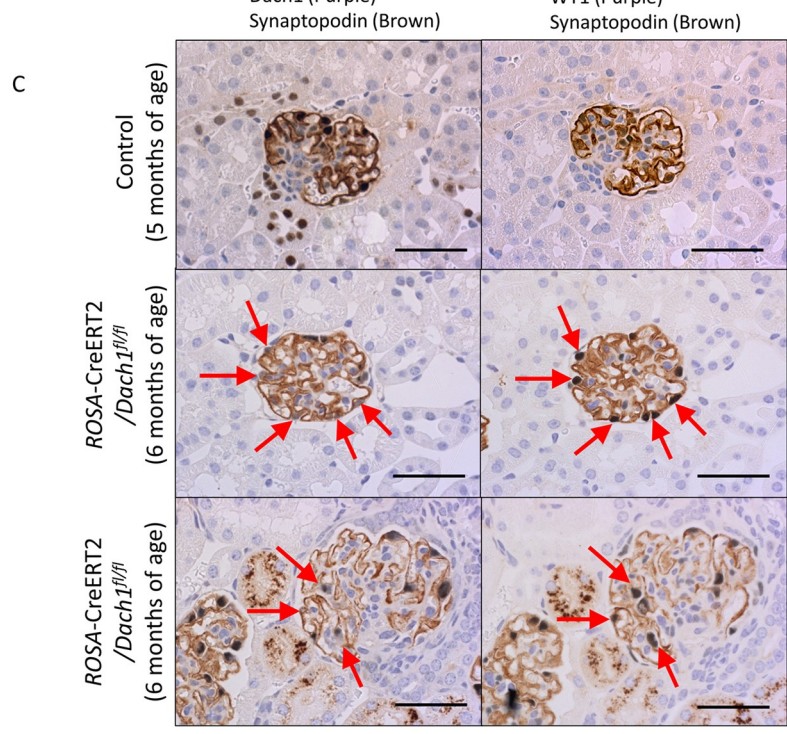

**Fig 3. Phenotypes of adult *ROSA*-CreERT2/*Dach1*<sup>fl/fl</sup> mice.** (A) Urinary albumin/creatinine ratio. *ROSA26*-CreERT2/*Dach1*<sup>fl/fl</sup> mice aged 15 weeks (n = 1), 18 weeks (n = 1), or 20–22 weeks (n = 10) were analyzed 3–5 days after the completion of three courses of tamoxifen treatment. Control mice aged 18 weeks (n = 1) or 20–22 weeks (n = 5) were similarly analyzed. Males are represented by black diamonds, and females are represented by white triangles. (B) PAS and nephrin staining in serial sections. *ROSA*-CreERT2/*Dach1*<sup>fl/fl</sup> mice exhibit segmental sclerosis with diminished nephrin staining (arrows). Scale bar: 100 μm. (C) Partial deletion of Dach1 in podocytes of *ROSA*-CreERT2/*Dach1*<sup>fl/fl</sup> mice. Serial sections were doubly stained for Dach1+ synaptopodin or WT1 + synaptopodin. Some podocytes in *ROSA*-CreERT2/*Dach1*<sup>fl/fl</sup> mice lacked Dach1 staining (arrows) but retained synaptopodin and WT1 staining (arrows).

Most Dach1 staining was absent in tubules in *ROSA*-CreERT2/*Dach1*<sup>fl/fl</sup> mice indicating efficient recombination in tubules. In the injured glomeruli of *ROSA*-CreERT2/*Dach1*<sup>fl/f</sup> mice (bottom), Dach1 and synaptopodin staining were diminished (arrows). Scale bar: 50 μm. Note: the secondary anti-mouse IgG antibody reacted with plasma components leaked into Bowman's capsule and reabsorbed in proximal tubules.

These findings suggest that Dach1 is essential for maintaining the normal integrity of podocytes in adult mice but does not affect the expression of distal tubule marker proteins.

## Discussion

Davis et al. (24) reported normal renal morphology in global *Dach1* knockout (KO) mice. Here, we explored the renal phenotypes of our *Dach1* KO line, finding largely normal morphology, consistent with Davis's findings but contrasting with Cao et al.'s observations (26). Cao's study revealed severe renal hypoplasia, reduced glomerular numbers, and immature podocytes in homozygous *Dach1* null mice. This variance prompts consideration of the differences in KO gene structures. In our global Dach1 KO mouse line, the deletion of the first exon encoding 275 amino acids, including the Dachshund motif-N, was noted. In Davis's *Dach1* KO mice, the first exon was replaced with a PGK-hprt cassette. In contrast, Cao's KO mice (*Dach1tm1a*) featured a long fragment insertion containing the lacZ gene and neomycin-resistant cassette in the first intron. Notably, the insertion of selection cassettes can dysregulate neighboring gene expression, as observed in other knockout models [31–33]. Thus, the disruption of the interaction between neighboring genes and regulatory elements by the inserted knockout-first cassette might explain the discrepancies observed [34, 35]. Investigating global KO mice carrying the allele with FLP and Cre recombination (*Dach1tm1d*) may shed light on this. Strain differences may also contribute; our *Dach1*<sup>-/-</sup> mice had a mixed genetic background of C57BL/6J and 129S6/SvEvTac, while Cao's KO mice were inbred C57BL/6 mice. Additionally, translation initiation from the second ATG codon in the second exon may yield a shorter protein (amino acids 298–751), which could partially account for the observed phenotypes in our and Davis's *Dach1* KO mice. However, we believe this is unlikely because our *Nphs1*-Cre/*Dach1*<sup>fl/fl</sup> mice lacking the first exon, exhibited more severe phenotypes at baseline than Cao's podocyte-specific *Dach1* KO mice lacking the second exon.

In a previous study by Endlich, the suppression of *Dachd*, the *Dach1* ortholog, in zebrafish resulted in the downregulation of nephrin, podocin, and synaptopodin, as well as the disruption of the filtration barrier [23]. Therefore, we conducted detailed analyses, expecting subtle abnormalities in the podocytes of global *Dach1* KO mice. However, our analyses revealed no abnormalities in the light microscopic morphology, ultrastructure, or expression of various podocyte marker proteins. Thus, the absence of Dach1 alone does not impact podocyte development. Since Dach2 is not expressed in podocytes as shown by our previous study (2) and other databases [36], it is unlikely that Dach2 can compensate for the lack of Dach1 (2). Our study demonstrated concentrated and intensified Dach1 staining in podocytes at the capillary loop stage but not in precursor cells at earlier stages, suggesting that Dach1 plays a more significant role in podocytes after differentiation is completed.

There is another disparity between Cao's findings and our findings in the baseline characteristics of podocyte-specific *Dach1* KO mice. In Cao's study, despite the nearly complete loss of Dach1 staining in podocytes, these mice displayed no proteinuria, maintained a normal glomerular structure, and retained an intact foot-process architecture in adulthood. However, severe podocyte damage ensued upon the induction of a diabetic state or treatment with adriamycin. In contrast, among our podocyte-specific *Dach1* KO mice, a

significant portion exhibited focal segmental glomerulosclerosis (FSGS) at baseline despite a low rate of Dach1 deletion. Similar observations were made with our inducible *Dach1* KO mice.

The reason for this discrepancy also remains unclear. Both Cao's and our podocyte-specific *Dach1* KO mice share the C57BL/6 genetic background. Our inducible *Dach1* KO mice had mixed genetic backgrounds, indicating that basal podocyte injury is not limited to a specific genetic background. The main difference is that the first exon is deleted in our mice, whereas it remains intact in Cao's podocyte-specific *Dach1* KO mice [26]. It is possible that the product of the first exon is expressed and partially functions to maintain normal architecture in the podocytes of Cao's mice. The product of the first exon is not recognized by the Dach1 antibody used in their study (Millipore Sigma HPA012672i). Finally, variances in microbiome status, nutritional status, or other environmental factors may induce subtle changes in podocyte protein expression, leading to discrepancies in phenotypes between studies, as observed in the patients with ApoL1 risk variants [37].

In both *Nphs1*-Cre/*Dach1fl/fl* and *Rosa*-Cre/*Dach1fl/fl* mice, Dach1-positive podocytes were rarely observed within the sclerotic lesions. This may challenge the causal role of Dach1 deficiency to glomerulosclerosis. We previously demonstrated that when a fraction of the podocytes is injured, other initially intact podocytes within the same glomerulus are secondarily injured and involved in sclerotic lesions [38, 39]. Thus, Dach1-positive podocytes do not lead to glomerulosclerosis but are involved in the sclerotic lesions caused by adjacent Dach1-deficient podocytes.

Due to the limited percentage of *Dach1* deleted podocytes in our mice, we did not perform further analyses, such as those elegantly demonstrated by Cao's group. Nevertheless, our study indicated that Dach1 deficiency alone, without any secondary hits, can induce podocyte injury in adult mice. This finding suggests that additional pathways beyond those identified in Cao's study are involved in the maintenance of the normal integrity of podocytes. Dach1 is downregulated in injured podocytes in patients with various kidney diseases, which may further facilitate podocyte injury. Further investigation of downstream molecules of Dach1 may provide insight into the pathogenesis of late-onset podocytopathies, such as aging kidneys.

In summary, our findings provide additional insights into the renal phenotypes of *Dach1* KO mice. We observed discrepancies between different *Dach1* KO models. Further studies are needed to elucidate the precise mechanisms underlying these discrepancies and the role of Dach1 in podocytes, which will contribute to our knowledge of renal pathophysiology and potentially open avenues for therapeutic interventions in kidney diseases.

## Supporting information

**S1 Fig. Immunostaining for nephrin, podocin, podocalyxin, synaptopodin, nestin and Dach1 in serial sections of *Dach1-/-* and littermate control mice.** There were no differences in these podocyte proteins. Scale bar: 100 μm.
(TIF)

**S2 Fig. Urinary albumin/creatinine (mg/mg) ratio in 7- to 8-month-old *Nphs1*-Cre/ *Dach1fl/fl* mice.** In S2 and S4 Figs, Males: black diamonds. Females: white triangles. For S4 Fig, The numbers of podocytes in intact glomeruli were counted in samples doubly stained for WT1 and synaptopodin. There was no difference in the average number of podocytes per glomerulus between *Nphs1*-Cre/*Dach1fl/fl* and control mice. Not significant (n.s.).
(TIF)

**S3 Fig. The SDS-PAGE pictures of the urines of *Nphs1*-Cre/*Dach1*<sup>fl/fl</sup> and control mice.** In S2 and S4 Figs, Males: black diamonds. Females: white triangles. For S4 Fig, The numbers of podocytes in intact glomeruli were counted in samples doubly stained for WT1 and synaptopodin. There was no difference in the average number of podocytes per glomerulus between *Nphs1*-Cre/*Dach1*<sup>fl/fl</sup> and control mice. Not significant (n.s.).
(TIF)

**S4 Fig. Number of podocytes per glomerulus in 7- to 8-month-old *Nphs1*-Cre/*Dach1*<sup>fl/fl</sup> mice.** In S2 and S4 Figs, Males: black diamonds. Females: white triangles. For S4 Fig, The numbers of podocytes in intact glomeruli were counted in samples doubly stained for WT1 and synaptopodin. There was no difference in the average number of podocytes per glomerulus between *Nphs1*-Cre/*Dach1*<sup>fl/fl</sup> and control mice. Not significant (n.s.).
(TIF)

**S5 Fig. Scanning electron microscopy of *Nphs1*-Cre/*Dach1*<sup>fl/fl</sup> mice.** Podocytes from *Nphs1*-Cre/*Dach1*<sup>fl/fl</sup> mice exhibited thickening of foot processes compared with control mice. Scale bar: 1 μm.
(TIF)

**S6 Fig. PAS and nephrin staining in a biopsied *ROSA*-CreERT2/*Dach1*<sup>fl/fl</sup> mouse.** *ROSA*-CreERT2/*Dach1*<sup>fl/fl</sup> mouse exhibited normal histology with normal nephrin staining one week after the end of tamoxifen treatment at 18 weeks of age, but the same mouse showed segmental sclerosis with diminished nephrin staining (arrows) 8 weeks later (26 weeks of age). Scale bar: 50 μm.
(TIF)

**S7 Fig. Scanning electron microscopy of *ROSA*-CreERT2/*Dach1*<sup>fl/fl</sup> mice.** Podocytes of *ROSA*-CreERT2/*Dach1*<sup>fl/fl</sup> mice exhibited thickening of the foot processes compared to those of the control. Scale bar: 1 μm.
(TIF)

**S8 Fig. Immunostaining for Dach1 and uromodulin, NCC, calbindin-D-28K in serial sections of *ROSA*-CreERT2/*Dach1*<sup>fl/fl</sup> mice.** Dach1 staining was retained in glomeruli due to inefficient Cre-mediated recombination in podocytes. Although most of the tubular Dach1 staining disappeared, no abnormalities were observed in the intensity or pattern of the staining of uromodulin, NCC, calbindin-D-28K. Calbindin-D-28K is highly expressed in DCT and CNT, and weakly expressed in CCD. Information about the primary antibodies, antigen retrieval methods, and dilution ratios are listed in the S1 Table. Scale bar: 100 μm. S8 Fig. Immunostaining for Dach1 and α, β, γ ENaCs, or AQP2 in serial sections of *ROSA*-CreERT2/*Dach1*<sup>fl/fl</sup> mice. No abnormalities were observed in the intensity or pattern of the staining of α, β, γ ENaCs, or AQP2 in the kidney lacking Dach1. Information about the primary antibodies, antigen retrieval methods, and dilution ratios are listed in the S1 Table. Scale bar: 100 μm.
(TIF)

**S1 Table. Antibodies used for immunostaining.**
(TIF)

**S2 Table. Raw data for Figs 1–3.**
(XLSX)

**S1 File. Original images.** Original images for S3 Fig.
(PDF)

## Acknowledgments

We acknowledge Ms. Shiho Imai, Ms. Chie Sakurai and the Support Center for Medical Research and Education of Tokai University for excellent technical assistance and Ms. Yukiko Tanaka for administrative assistance.

## Author Contributions

**Conceptualization:** Taiji Matsusaka.

**Funding acquisition:** Taiji Matsusaka.

**Investigation:** Keiko Tanaka, Haruko Hayasaka, Taiji Matsusaka.

**Methodology:** Keiko Tanaka, Taiji Matsusaka.

**Project administration:** Taiji Matsusaka.

**Resources:** Haruko Hayasaka, Taiji Matsusaka.

**Supervision:** Taiji Matsusaka.

**Validation:** Haruko Hayasaka, Taiji Matsusaka.

**Visualization:** Keiko Tanaka, Taiji Matsusaka.

**Writing – original draft:** Keiko Tanaka, Taiji Matsusaka.

**Writing – review & editing:** Haruko Hayasaka, Taiji Matsusaka.

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
