## [Decision Letter · Decision Letter 0]

8 Jan 2024

PONE-D-23-40891Dach1 is essential for maintaining normal mature podocytesPLOS ONE

Dear Dr. Matsusaka,

Thank you for submitting your manuscript to PLOS ONE. After careful consideration, we feel that it has merit but does not fully meet PLOS ONE’s publication criteria as it currently stands. Therefore, we invite you to submit a revised version of the manuscript that addresses the points raised during the review process.

We look forward to receiving your revised manuscript.

Kind regards,

Zhanjun Jia

Academic Editor

PLOS ONE

Journal Requirements:

Reviewers' comments:

Reviewer's Responses to Questions

**Comments to the Author**

1. Is the manuscript technically sound, and do the data support the conclusions?

Reviewer #1: Yes

Reviewer #2: Partly

Reviewer #3: Partly

Reviewer #4: Yes

2. Has the statistical analysis been performed appropriately and rigorously? 

Reviewer #1: Yes

Reviewer #2: No

Reviewer #3: I Don't Know

Reviewer #4: Yes

3. Have the authors made all data underlying the findings in their manuscript fully available?

Reviewer #1: Yes

Reviewer #2: Yes

Reviewer #3: Yes

Reviewer #4: Yes

4. Is the manuscript presented in an intelligible fashion and written in standard English?

Reviewer #1: Yes

Reviewer #2: Yes

Reviewer #3: Yes

Reviewer #4: Yes

5. Review Comments to the Author

Reviewer #1: This is a well-designed and interesting article regarding podocyte pathophisiology. I have several concerns which should be addressed adequately.

1. It is nice to examine some interactions between Dach1 and WT1 expression in the podocyte.

2. How about the expression of DACH1 is in renal biopsy specimens from patients with WT1 mutations?

3. The authors examined the implication of DACH1 in in vivo murine models. Were the results obtained same as those of in a cultured human podocyte?

4. The implication of the results obtained for future perspective particulary in clinical applications should be mentioned.

5. When podocytes injured, was the loss of DACA1 expression firster than that of the other sturauctual molecules?

Reviewer #2: Matsusaka et al present a manuscript describing the renal, specifically the glomerular, phenotype of several DACH1 knockout animals. DACH1 has previously been described by this group and others as a gene expressed highly in glomerular podocytes as well as other renal cells and non-renal tissues, with loss of DACH1 expression in injured glomeruli. Global DACH1 knockout has been reported to confer perinatal lethality by an unclear mechanism; renal hypoplasia was reported by one group, though its connection to lethality was unclear. Cau et al previously reported that podocyte specific knockout animals are normal into adulthood but are highly susceptible to glomerular stress, including diabetes. The current group, using a previously unreported knockout construct, report slightly smaller kidneys in the global DACH1 knockout, but no loss of glomerular number or architecture. Animals with a podocyte-specific Cre driven knockout show overall poor Cre-mediated excision in podocytes, and a variable glomerular phenotype with albuminuria and glomerular scerlosis. Global DACH1 knockout during adulthood also leads to only incomplete DACH1 loss in glomeruli, and a mild renal phenotype. Overall, the results are of interest to the glomerular disease research community, but the results are difficult to fully interpret given their presentation, and the analysis is rather superficial and limited. Clearer presentation of the results already at hand is needed, though additional data generation would be of benefit.

Major concerns:

1. Histologic analysis is provided for 11 podocyte-specific knockout mice. However, per methods, these span in age from 8 weeks to 7-8 months. The age of control mice is not provided. Given the wide span in ages and the authors’ report that albuminuria worsens with age, combining all age groups for analysis is no appropriate. As the authors report that 7 experimental animals were of age 7-8 months, it would seem reasonable to limit histologic analysis to this group, comparing them to similarly aged control mice. Sex-specific data also need to be provided, as there are many instances of sex-specific differences in susceptibility to glomerular injury.

2. The degree of DACH1 excision in podocytes of the NPHS1-Cre and the ROSA-Cre mice is difficult to gauge from the available images and data provided by the authors. It would be helpful to provide some quantitative analysis, or co-immunofluorescence staining with WT1, of a young (perhaps 1mo old) animal to better gauge the efficacy of Cre-mediated excision. An analysis of total podocyte number might also be informative, as one could postulate that once DACH1 is lost in podocytes, these particular cells rapidly die and are therefore not seen in IHC analysis.

3. It would be helpful to provide experimental details for the tamoxifen-mediated DACH1 knockout experiments in the methods, or at the beginning of the results section related to these experiments. Specifically, please specify the age at which Cre-induction occurred, the time between tamoxifen administration and urine and histological analysis, and the sex of the animals.

4. The authors conclude that the lack of staining of several distal nephron markers is indicative of no suggestion of a critical role for DACH1 in tubular cells. This seems a premature conclusion. Urine concentrating ability, assessment of serum electrolyte and acid-base parameters would seem to be useful in better addressing whether there are functional effects on the distal nephron. But even here, it would be possible to postulate indirect effects from other organ systems affected by global DACH knockout. It may be best to temper the language of the conclusion, and just state that distal nephron marker expression was not affected in these animals.

5. Addition of some mechanistic insight into what role DACH1 is playing in the maintenance of podocytes and their function would really add to the manuscript. While not absolutely necessary, without mechanistic data, the descriptive analysis of the renal pathology needs to be enhanced. This could include glomerular injury scores across time and between sexes, analysis of podocyte structure by ultrastructure beyond the single SEM image shown for the NPHS1-Cre;DACH1 flox mice.

Minor:

There is mention of performing a renal biopsy on a mouse in the results section. This is not a routine procedure performed on mice in this reviewer’s experience; please provide details in the methods section. In addition, it would seem that a renal biopsy/partial nephrectomy in a mouse might lead to proteinuria and/or FSGS lesions afterwards, or at least act as a potential second hit. It is not clear that providing histology from this animal (supplemental figure 2) is helpful in assessing the development of pathology in the tamoxifen induced knockout animals. More granular data presentations, as noted above, would be more useful overall.

Reviewer #3: In this manuscript Tanaka and colleagues analyze the phenotypic effects of global, tissue-specific and temporal deletion of Dach1 expression in mice. They find that ubiquitous deletion leads to perinatal death consistent with two previous reports but, in contrast to one previous study, they find relatively normal podocyte and kidney development and conclude that Dach1 is not essential for podocyte precursor differentiation. More strikingly, in adult mice they find that podocyte specific KO of Dach1 (as well as ubiquitous inducible deletion in adult) leads to mice with progressive albuminuria and an FSGS-like syndrome while previous studies have suggested virtually no baseline phenotype in similar KOs (although the latter mice do show a potent susceptibility to injury induced FSGS-like disease). They speculate that these differences could reflect differences in the architecture of the constructs used to generate the KOs and highlight the need for further studies to elucidate the precise reasons for these discrepancies.

In general I find these studies to well-performed and detailed but the conclusions are somewhat dissatisfying in that the authors have not been able to identify a clear reason for the discrepancies between their studies and the previous ones. While some of reported differences appear large and noteworthy, they may not be as significant as the authors conclude. I think they could consider and discuss a much larger range of potential causes for the discrepancy in results between their studies and those of other groups. For example:

1) One rather trivial explanation for the differences between the complete KO data could be strain differences. I note that the authors here have used ES cells from “C57BL/6J x 129S6/SvEvTac F1 mice” to make there KO and then sibling crossed these mice to generate homozygotes. This hybrid background, rather than the cited differences in genetic constructs, is a second possible cause for the discrepant results in the perinatal studies.

2) With regard to the phenotypes in the adult ubiquitous and podocyte-specific deletions, the authors identify what appears to be a more striking phenotype in that their mice exhibit clear evidence of an FSGS-like disease progression whereas previous groups observed normal phenotypes unless the mice were challenged pharmacologically. But this too may actually be a more subtle difference. For example, it is known that a mere 50% reduction in expression of the major podocyte sialomucin, podocalyxin, renders mice and humans highly susceptible to late onset FSGS-like syndromes (Refaeli I et al Sci. Reports 10: 9419, 2020, Barua M et al Kid Int. 85: 124, 2014, Lin FG et al Clin. Sci. 133: 9–21, 2018). This is one of many examples where a subtle change in protein expression can have major consequences on podocyte and kidney function. Accordingly, if in the current studies Dach1 deletion conferred susceptibility, but microbiome status, nutritional or other environmental influences that varied between studies altered the expression of other podocyte proteins, these very subtle differences between experimental conditions could account for what appear to be more major discrepancies between the studies. Minimally the authors should raise this as a possibility in their Discussion section.

Minor criticisms:

1) I was puzzled by thickened endothelial walls and the lack of obvious endothelia fenestrae in the TEM images in Fig 1. These should be readily apparent in glomerular endothelia. Is there an endothelial phenotype in the mice?

2) In Fig 2A there appears to be some evidence of weak deletion of the Dach1 targeted allele in the tail DNA of one of the samples. Is the NHPS1-Cre “leaky” and perhaps deleting in vascular endothelia or other cells in the tail?

3) The authors refer to inefficient deletion of Dach1 in podocytes based on residual expression in many of their mice. Can the authors be sure this inefficient deletion or does this mere reflect a high degree of protein stability that fails to disappear after gene deletion?

Reviewer #4: Tanaka et al. have conducted research on the function of Dach1 using various knockout (KO) mouse models. They have clearly elucidated the phenotypic differences among several types of Dach1 KO mice and highlighted the significance of Dach1. However, there are several concerns within this manuscript that need to be addressed.

Major

1. In Fig 1 Global Dach1 KO Mice, the authors report no significant differences in the expression of podocyte marker proteins between control and Dach1 KO mice in lines 171-173. However, there appear to be no images examining the expression of podocyte markers in this article. Given that the article's central theme is Dach1 and podocytes, the authors should include images that evaluate the podocyte markers.

2. The authors note that kidneys in global Dach1 KO mice were significantly smaller, yet the glomerular diameter and density were not reduced. However, they conclude that the absence of Dach1 causes kidney hypoplasia. Does this imply that Dach1 KO primarily affects tubules and collecting ducts formation without influencing the glomeruli? The mechanism behind this size difference should be explained in more detail.

3. In Fig 2 Nphs1-Cre/Dach1fl/fl mice, while Fig 2B shows the significance of albuminuria, the age of the mice evaluated is unclear.

4. Indeed, Fig 2C demonstrates the increased albuminuria in Nphs1-Cre/Dach1fl/fl mice as they age. However, it is unusual to show images of SDS-PAGE directly. Instead, the authors should present the quantified values of albuminuria by ELISA using a time-course figure, while the SDS-PAGE data could be included in a supplementary figure.

5. The results in Fig 2F are quite complex. Nphs1-Cre/Dach1fl/fl mice exhibited glomerulosclerosis and albuminuria, but the bottom figures of Fig 2F suggest that even preserved Dach1 expression might lead to glomerulosclerosis. Does this indicate that Dach1 expression is not directly associated with glomerulosclerosis? These results seem inconsistent with the authors' conclusion in this section.

6. In Fig 3 Inducible Dach1 KO Mice, similar to Fig 2B, the age of the mice evaluated in Fig 3A is unclear.

7. Also, like Fig 2F, Fig 3C suggests that Dach1 expression in podocytes does not influence podocyte survival. Unfortunately, these results do not support the authors' hypothesis.

8. In Disucssion, the detailed comparisons between the authors' data and previous studies are valuable. If possible, shorten the length of these comparisons.

9. The authors report that various types of Dach1 KO mice exhibit glomerulosclerosis and albuminuria. These findings are of significant interest and represent a novel contribution. However, some results seem inconsistent with the authors' hypothesis.

10. Both Table 1 and supplementary Table 1 are titled "Antibodies for Immunostaining." However, the distinction between these tables is unclear, even though they list different molecules. It is recommended to combine these tables into a single table, preferably as supplementary Table 1, for better clarity.

6. PLOS authors have the option to publish the peer review history of their article (what does this mean?). If published, this will include your full peer review and any attached files.

Reviewer #1: No

Reviewer #2: No

Reviewer #3: No

Reviewer #4: No

---

## [Author Response · Author response to Decision Letter 0]

28 Mar 2024

We thank you and the reviewers for their positive and encouraging evaluation of our manuscript, as well as for their valuable comments, which have helped us improve this paper. We performed additional analyses and experiments, including database analysis of the interaction between WT1 and Dach1, quantification of podocyte numbers in NPHS1-Cre/Dach1fl/fl and control mice, PCR analysis of the tail DNA of Nphs1-Cre/Dach1fl/wild mice, and immunostaining of global Dach1 KO and control mice.

Furthermore, we have added detailed experimental methods and information about mouse age and sex and thoroughly revised the manuscript, figures, and supplements. 

Point-by-point response to Reviewer 1's comments.

1. It is nice to examine some interactions between Dach1 and WT1 expression in the podocyte.

Response: Dach1 is known to form a transcriptional network with Six1 and Eya1,etc. Our previous study and other studies indicated that Dach1 and Wt1 are similarly suppressed in the injured glomeruli of individuals with various kidney diseases, which prompted us to investigate whether DACH1 and WT1 directly interact in podocytes. Through a survey of the STRING network (STRING-DQ.org), we found no significant evidence of physical or functional interactions between DACH1 and WT1. Notably, a prior study demonstrated that the WT1 protein binds to the Dach1 gene in mouse podocytes, although Dach1 gene expression did not significantly change with Wt1 gene knockout (J Am Soc Nephrol. 2015, 26:2118-28). Additionally, ChIP-Atlas data indicated that DACH1 binds to the WT1 gene in leukemia cells. However, data from Cao’s study (GSE16876) showed no alteration in Wt1 expression in Dach1 knockout podocytes, consistent with our observations of normal WT1 expression in podocytes of newborn global Dach1 KO mice. Thus, while both DACH1 and WT1 exhibit decreased expression in injured podocytes, the current data do not support a strong direct interaction. We have briefly mentioned this interaction in the introduction section (Page 2, Line 69-70).

2. How about the expression of DACH1 in renal biopsy specimens from patients with WT1 mutations?

Response: We currently lack access to human biopsy specimens harboring WT1 mutations. WT1 mutations typically result in Frasier and Denys–Drash syndromes, as well as sporadic FSGS. In podocytes affected by these conditions, it is conceivable that DACH1 expression is decreased, similar to what is observed in primary FSGS. Examination of RNA-seq data from Mariani Nephrotic Syndrome Glom via Nephroseq (https://nephroseq.org) indicated lower glomerular Dach1 mRNA levels in FSGS patients with proteinuria than in those without proteinuria. There are no reports on DACH1 expression in human podocytes with WT1 mutations. Based on findings in Wt1 knockout mice (J Am Soc Nephrol. 2015, 26:2118-28), it seems plausible that while Dach1 expression may not be directly regulated by WT1, its suppression could occur in injured podocytes due to WT1 mutation.

3. The authors examined the implication of DACH1 in in vivo murine models. Were the results obtained the same as those of a cultured human podocyte?

Response: Dach1 gene expression is markedly decreased in cultured podocytes. DACH1 expression is not detected or is very low in human podocyte cell lines, as evidenced by a recent comprehensive transcriptomic analysis (Kidney Int Rep. 2022, 8: 164-178). The functional implications of DACH1 in human podocyte cell lines remain unexplored. In our previous study, we observed attenuation of Dpp4 and Podxl1 mRNA expression upon Dach1 knockdown in primary cultured mouse podocytes. However, in the present study, mRNA analyses were not feasible due to the low rate of Cre recombination in podocytes.

4. The implication of the results obtained for future perspective, particularly in clinical applications should be mentioned.

Response: We and others have demonstrated the downregulation of Dach1 in injured podocytes in diverse kidney pathologies (Refs. 2, 22). Our study indicated that this downregulation may exacerbate podocyte injury. Exploring the downstream molecules influenced by Dach1 could shed light on the pathogenesis of late-onset podocytopathies, including those associated with aging kidneys. Following the reviewer's guidance, we have incorporated a discussion on the potential clinical implications of our findings (Page 12, Line 400-403).

5. When podocytes were injured, was the loss of DACA1 expression faster than that of the other structural molecules?

Response: Our results indicated that the absence of Dach1 compromises the maintenance of mature podocytes without requiring a secondary insult. One plausible mechanism is that Dach1 may play a vital role in regulating the expression of structural proteins that counteract filtration pressure, as suggested by the reviewer. Upon reviewing our previous microarray data from mouse-injured podocytes (Ref 2, Am J Physiol Renal Physiol 316: F241–F252, 2019), we observed a 0.71-fold decrease in Dach1 mRNA levels as early as 4 days after podocyte injury induction. This time course closely mirrors that of Nphs1 and Podxl1, which are typical podocyte-specific genes. Among the genes encoding structural proteins, 36 were substantially expressed in podocytes, of which six (Col4a3, Col18a1, Krt10, Krt81, Myl6b, and Sptbn1) exhibited similar downregulation patterns to Dach1 following injury induction. Additionally, we analyzed RNA-seq data from human FSGS glomeruli (Mariani Nephrotic Syndrome Glom), focusing on genes associated with structural molecules, such as ACTN4, ACTG1, COL4A3, COL4A4, COL4A5, MYL7, TUBB2B, and VIM, as their expression is notably concentrated in podocytes within glomeruli. The expression of these mRNAs was not lower in the glomeruli of proteinuric FSGS patients than in those of nonproteinuric FSGS patients. However, it remains uncertain whether structural protein genes are direct targets of Dach1. We briefly described the importance of elucidating Dach1 target genes in the Discussion section of the revised manuscript (Page 12, Line 401-403).

Point-by-point response to Reviewer 2's comments.

Major concerns:

1. Histologic analysis is provided for 11 podocyte-specific knockout mice. However, per methods, these span in age from 8 weeks to 7-8 months. The age of the control mice is not provided. Given the wide span in ages and the authors’ report that albuminuria worsens with age, combining all age groups for analysis is not appropriate. As the authors report that 7 experimental animals were of age 7-8 months, it would seem reasonable to limit histologic analysis to this group, comparing them to similarly aged control mice. Sex-specific data also need to be provided, as there are many instances of sex-specific differences in susceptibility to glomerular injury.

Response: We have included the age and sex details of the control mice in the Methods section, along with the corresponding figures. In accordance with the reviewer's suggestion, we have added the FSGS ratio and average urinary albumin creatinine ratio restricted to mice aged 7-8 months, as noted in the Results section (Page 7, Line 224-225 and Line 228-229) and Supplementary Figure 2. However, due to the limited number of female Nphs1-Cre/Dach1fl/fl mice available for analysis (only three), statistical analysis specific to sex was not feasible. Nevertheless, the sex distribution is illustrated in the updated Figure 2B and 2E.

2. The degree of DACH1 excision in podocytes of the NPHS1-Cre and the ROSA-Cre mice is difficult to gauge from the available images and data provided by the authors. It would be helpful to provide some quantitative analysis, or co-immunofluorescence staining with WT1, of a young (perhaps 1mo old) animal to better gauge the efficacy of Cre-mediated excision. An analysis of total podocyte number might also be informative, as one could postulate that once DACH1 is lost in podocytes, these particular cells rapidly die and are therefore not seen in IHC analysis.

Response: As the reviewer suggested, if Dach1 KO podocytes are lost without causing injury, the Dach1 KO ratio and the impact of Dach1 KO are underestimated. We conducted double staining for WT1 and synaptopodin in Nphs1-Cre/Dach1fl/fl and control mice at 7-8 months of age. We quantified the number of WT1+ synaptopodin+ podocytes in uninjured glomeruli and found no significant difference in the number of podocytes between the KO and control mice. This finding suggested that Dach1 KO podocytes are not lost without causing injury, thereby ruling out underestimation of the Dach1 KO ratio and its impact. The data illustrating this observation have been incorporated into the Results section (Page 7, Line 241-247) and the new Figure S4.

We performed co-immunofluorescence staining of Dach1 and WT1 in paraffin samples using a FlexAble labeling kit (Proteintech). However, due to high autofluorescence in the paraffin sections, we were unable to detect the staining effectively. Although precise quantification of Dach1 KO in serial section analyses remains challenging, the percentage of Dach1 KO podocytes in the podocytes of Nphs1-Cre/Dach1fl/fl mice was less than 10%. This information has been included in the Results section (Page 7, Line 238-239).

3. It would be helpful to provide experimental details for the tamoxifen-mediated DACH1 knockout experiments in the methods, or at the beginning of the results section related to these experiments. Specifically, please specify the age at which Cre-induction occurred, the time between tamoxifen administration and urine and histological analysis, and the sex of the animals.

Response: We acknowledge the reviewer's comment regarding the need for explicit experimental details about the tamoxifen-mediated Dach1 knockout experiments. We treated Rosa-Cre/Dachfl/fl mice with tamoxifen for a total of 15 days, beginning at ages ranging from 4 to 11 weeks. After the completion of tamoxifen treatment, we collected 24-hour urine samples within a window of 3-5 days. Except for one mouse, all subjects were euthanized within 3-5 days after treatment completion. Notably, an additional Nphs1-Cre/Dachfl/fl mouse, along with a control counterpart, underwent renal biopsy (partial nephrectomy) 3 days following the completion of tamoxifen treatment and was then sacrificed 60 days later. We have diligently incorporated these critical details regarding the timing of tamoxifen administration, analyses, and sex of the mice into the Methods section (Page 4, Line 116-124). The sex distribution is illustrated in the updated Figure 3A.

4. The authors conclude that the lack of staining of several distal nephron markers is indicative of no suggestion of a critical role for DACH1 in tubular cells. This seems a premature conclusion. Urine concentrating ability, and assessment of serum electrolyte and acid-base parameters would seem to be useful in better addressing whether there are functional effects on the distal nephron. But even here, it would be possible to postulate indirect effects from other organ systems affected by global DACH knockout. It may be best to temper the language of the conclusion, and just state that distal nephron marker expression was not affected in these animals.

Response: We concur with the reviewer's assessment. Our study did not specifically investigate tubular functions. Accordingly, we have revised the final sentence of the Results section (Page 9, Line 310-312) per the reviewer's suggestion.

5. The addition of some mechanistic insight into what role DACH1 is playing in the maintenance of podocytes and their function would really add to the manuscript. While not absolutely necessary, without mechanistic data, the descriptive analysis of the renal pathology needs to be enhanced. This could include glomerular injury scores across time and between sexes, and analysis of podocyte structure by ultrastructure beyond the single SEM image shown for the Nphs1-Cre;DACH1 flox mice.

Response: We acknowledge the limitations highlighted by the reviewer regarding the lack of mechanistic insight in our study. Due to the low Cre recombination rate, we were unable to conduct a comprehensive molecular characterization of Dach1 KO podocytes. Nonetheless, our findings suggest that Dach1 deficiency alone can disrupt the maintenance of normal adult podocytes but not the differentiation of cells into podocytes during development. We hypothesize the existence of additional downstream pathways beyond those identified in Cao's study. This speculation is briefly discussed in the Discussion section (Page 12, Line 398-403).

Minor:

There is mention of performing a renal biopsy on a mouse in the results section. This is not a routine procedure performed on mice in this reviewer’s experience; please provide details in the methods section. In addition, it would seem that a renal biopsy/partial nephrectomy in a mouse might lead to proteinuria and/or FSGS lesions afterwards, or at least act as a potential second hit. It is not clear that providing histology from this animal (supplemental figure 2) is helpful in assessing the development of pathology in the tamoxifen induced knockout animals. More granular data presentations, as noted above, would be more useful overall.

Response: The renal biopsy procedure has been included in the Methods section (Page 5, Line 152-158). Unlike rats, mice typically exhibit resistance to nephron reduction. Most mouse strains do not exhibit FSGS even following subtotal nephrectomy. Following biopsy, less than 20% of the kidney volume (i.e., less than 10% of the total nephron number) was lost by excision and coagulation. Furthermore, the control mouse that underwent a similar renal biopsy exhibited normal findings. Therefore, we posit that the biopsy did not induce FSGS. We euthanized all other ROSA-Cre/Dach1fl/fl mice 3-5 days after completing the three courses of tamoxifen treatment. The observation of the biopsied mouse implies the potential progression of FSGS lesions over time after tamoxifen treatment. It is conceivable that if we had assessed ROSA-Cre/Dach1fl/fl mice at a later time point, the prevalence of FSGS would likely have been greater. However, this speculation is based on the findings of a single mouse, which we did not emphasize.

Point-by-point response to Reviewer 3's comments.

1) One rather trivial explanation for the differences between the complete KO data could be strain differences. I note that the authors here have used ES cells from “C57BL/6J x 129S6/SvEvTac F1 mice” to make there KO and then sibling crossed these mice to generate homozygotes. This hybrid background, rather than the cited differences in genetic constructs, is a second possible cause for the discrepant results in the perinatal studies.

Response: We concur with the Reviewer regarding the potential impact of genetic background. Our Dach1-/- mice were bred on a mixed genetic background, while those utilized in Dr. Cao’s study were bred on an inbred C57BL/6 genetic background. The pure C57BL/6 genetic background may have induced significantly abnormal phenotypes in global Dach1 KO mice, although this does not fully explain the variance in basal phenotypes observed in podocyte-specific Dach1 KO mice.

We have incorporated this possibility into the Discussion section (Page 10, Line 347-349).

2) With regard to the phenotypes in the adult ubiquitous and podocyte-specific deletions, the authors identify what appears to be a more striking phenotype in that their mice exhibit clear evidence of an FSGS-like disease progression whereas previous groups observed normal phenotypes unless the mice were challenged pharmacologically. But this too may actually be a more subtle difference. For example, it is known that a mere 50% reduction in expression of the major podocyte sialomucin, podocalyxin, renders mice and humans highly susceptible to late onset FSGS-like syndromes (Refaeli I et al Sci. Reports 10: 9419, 2020, Barua M et al Kid Int. 85: 124, 2014, Lin FG et al Clin. Sci. 133: 9–21, 2018). This is one of many examples where a subtle change in protein expression can have major consequences on podocyte and kidney function. Accordingly, if in the current studies Dach1 deletion conferred susceptibility, but microbiome status, nutritional or other environmen

---

## [Decision Letter · Decision Letter 1]

3 May 2024

Dach1 is essential for maintaining normal mature podocytes

PONE-D-23-40891R1

Dear Dr. Matsusaka,

We’re pleased to inform you that your manuscript has been judged scientifically suitable for publication and will be formally accepted for publication once it meets all outstanding technical requirements.

Kind regards,

Zhanjun Jia

Academic Editor

PLOS ONE

Additional Editor Comments (optional):

Reviewers' comments:

Reviewer's Responses to Questions

**Comments to the Author**

1. If the authors have adequately addressed your comments raised in a previous round of review and you feel that this manuscript is now acceptable for publication, you may indicate that here to bypass the “Comments to the Author” section, enter your conflict of interest statement in the “Confidential to Editor” section, and submit your "Accept" recommendation.

Reviewer #1: All comments have been addressed

Reviewer #3: All comments have been addressed

2. Is the manuscript technically sound, and do the data support the conclusions?

Reviewer #1: Yes

Reviewer #3: Yes

3. Has the statistical analysis been performed appropriately and rigorously? 

Reviewer #1: Yes

Reviewer #3: Yes

4. Have the authors made all data underlying the findings in their manuscript fully available?

Reviewer #1: Yes

Reviewer #3: Yes

5. Is the manuscript presented in an intelligible fashion and written in standard English?

Reviewer #1: Yes

Reviewer #3: Yes

6. Review Comments to the Author

Reviewer #1: The authors addressed their original MS. The revised MS is addressed almost adequately. I have no further concerns.

Reviewer #3: (No Response)

7. PLOS authors have the option to publish the peer review history of their article (what does this mean?). If published, this will include your full peer review and any attached files.

Reviewer #1: **Yes: **Hiroshi Tanaka

Reviewer #3: No

---

## [Editor Report · Acceptance letter]

15 May 2024

PONE-D-23-40891R1 

PLOS ONE

Dear Dr. Matsusaka, 

I'm pleased to inform you that your manuscript has been deemed suitable for publication in PLOS ONE. Congratulations! Your manuscript is now being handed over to our production team.

Kind regards, 

on behalf of

Dr. Zhanjun Jia 

Academic Editor

PLOS ONE